# Dolphin Pituitary Gland: Immunohistochemistry and Ultrastructural Cell Characterization Following a Novel Anatomical Dissection Protocol and Non-Invasive Imaging (MRI)

**DOI:** 10.3390/ani15050735

**Published:** 2025-03-04

**Authors:** Paula Alonso-Almorox, Alfonso Blanco, Carla Fiorito, Eva Sierra, Cristian Suárez-Santana, Francesco Consolli, Manuel Arbelo, Raiden Grandía Guzmán, Ignacio Molpeceres-Diego, Antonio Fernández Gómez, Javier Almunia, Ayoze Castro-Alonso, Antonio Fernández

**Affiliations:** 1Veterinary Histology and Pathology, Atlantic Center for Cetacean Research (CAIC), Institute of Animal Health and Food Safety (IUSA), Veterinary School, University of Las Palmas de Gran Canaria (ULPGC), Trasmontaña s/n, 35413 Arucas, Spain; eva.sierra@ulpgc.es (E.S.); cristian.suarez@ulpgc.es (C.S.-S.); francesco.consoli@unich.it (F.C.); manuel.arbelo@ulpgc.es (M.A.); raiden.grandia@gmail.com (R.G.G.); ignamolp11@gmail.com (I.M.-D.); ayoze.castro@ulpgc.es (A.C.-A.); antonio.fernandez@ulpgc.es (A.F.); 2Department of Anatomy and Comparative Pathology and Anatomy, University of Cordoba, 14014 Cordoba, Spain; an1brloa@uco.es; 3Centro para el Estudio de Sistemas Marinos CESIMAR-CONICET, Puerto Madryn 9120, Argentina; carlafiorito@gmail.com; 4Servicio de Anatomía Patológica, Hospital Universitario de Gran Canaria Dr. Negrín, 35010 Las Palmas, Spain; antoniofergom@hotmail.com; 5Loro Parque Fundación, Avda. Loro Parque s/n, 38400 Puerto de la Cruz, Spain; dir@loroparque-fundacion.org

**Keywords:** pituitary, adenohypophysis, neurohypophysis, odontocetes, comparative anatomy, morphology, ultrastructure

## Abstract

The pituitary gland is a vital organ in mammals that helps regulate essential bodily functions, but our understanding of this organ in dolphins is limited. This study investigated the structure and function of the dolphin pituitary gland using advanced imaging, dissection techniques, and microscopic analysis. We examined pituitary glands from three dolphin species and identified eight types of cells responsible for producing important hormones that control growth, reproduction, metabolism, and stress responses. For the first time, we could stain thyrotrophs with the corresponding antibodies, and analyzed these cells in detail with high-resolution microscopy, providing a clearer picture of their role in dolphin health. This research improves our understanding of dolphin biology and has practical benefits for their care, conservation, and welfare, especially as these species face increasing environmental challenges.

## 1. Introduction

The pituitary gland is a key endocrine organ that serves as the central intermediary for numerous critical nervous, endocrine, and humoral physiological functions [1]. Despite its reputation as the “master gland” and its essential role in the neuroendocrine system’s central axis, research on the pituitary gland in dolphins remains limited.

In cetaceans, the gland is a bean-shaped structure, generally wider than it is deep, located in a shallow depression of the sphenoid bone at the skull’s base. This anatomical area, known as the “*sella turcica*” in most mammals, is either absent or modified in most cetaceans [2,3].

The gland was first described in odontocetes by Wislocky in 1929 in the Atlantic bottlenose dolphin (*Tursiops truncatus*). Most subsequent morphological studies on cetaceans were conducted during the 1950s and 1960s [3,4,5,6], coinciding with the peak of whaling activities, with only a few more recent contributions on morphology or function [2,7,8].

The pituitary gland in cetaceans is composed of two main regions: the adenohypophysis (glandular portion) and the neurohypophysis (neural portion or *pars nervosa*). The adenohypophysis is further divided into the *pars distalis*, which constitutes the body of the gland and is highly vascularized, and the *pars tuberalis,* which surrounds the infundibular stalk and is also richly supplied by a capillary network [2].

In cetaceans, the adenohypophysis and neurohypophysis are fully separated and individually encapsulated by the dura mater, with an opening that allows the hypophyseal stalk to connect to the hypothalamus [9,10]. This differs strikingly from most mammals, where a middle portion (*pars intermedia)* juxtaposes these structures (Appendix A) [11,12]. The presence and differentiation of the *pars intermedia* in cetaceans remain inconsistent and poorly characterized. Some studies suggest its complete absence in certain cetacean species due to the lack of embryological contact between the *pars distalis* and *pars nervosa* during development [8,9,13], while others identify a distinct, organized area in the dorsal aspect of the adenohypophysis, potentially representing a remnant of the *pars intermedia* [2].

Histologically, the cetacean pituitary gland closely resembles that of other mammals. In the *pars distalis*, the parenchyma is arranged into irregular cords or small clusters of endocrine cells, forming a dense and organized network. These cells are closely associated with sinusoidal fenestrated sinusoidal capillaries, allowing the diffusion of hormones into the bloodstream [2]. Immunohistochemical studies have identified the two known types of acidophilic cells (somatotrophs and lactotrophs) in cetaceans [2,14,15]. However, among the four recognized basophilic cell types, only corticotrophs, gonadotrophs, and melanotrophs have been detected, while thyrotrophs have yet to be immunolabeled in these species. Electron microscopy has revealed six distinct cell types in cetaceans [16]. One classification system, based on the morphology of cytoplasmatic granules, was proposed, although it did not specifically identify the endocrine cell types [17].

On the other hand, the neurohypophysis of cetaceans has received even less attention. Only a few publications have reported detailed anatomical dissections and histological features [2,7]. Extracting the entire gland to adequately study the cytohistological characteristics of the neurohypophysis remains a significant challenge.

In this study, we present an innovative methodology for skull opening, incorporating non-invasive imaging technology before necropsy. This approach allows for the complete extraction of the pituitary gland, along with its neurohistological connection to the hypothalamus. Additionally, given the scarcity of immunohistochemical and ultrastructural studies on the cetacean pituitary gland, we contribute to the identification of adenohypophyseal cells using immunomarkers in three dolphin species—the common dolphin (*Delphinus delphis*), the common bottlenose dolphin (*Tursiops truncatus*), and the Atlantic spotted dolphin (*Stenella frontalis)*. We also provide a detailed ultrastructural characterization of each adenohypophyseal cell in common dolphins.

## 2. Materials and Methods

### 2.1. Animals

We collected the pituitary gland from 47 dolphins (*n* = 47) that had either died in facilities under human care (Loro Parque, Canary Islands, Spain) or stranded within the Canary Archipelago between 2021 and 2024. The animals underwent full post-mortem exams, performed within 24 h postmortem (hpm) by veterinarians from the pathology team at the Animal Health and Food Safety Institute (IUSA), Faculty of Veterinary Medicine, Universidad de Las Palmas de Gran Canaria. We followed the basic necropsy protocol outlined by Kuiken and Garcia Hartmann [18], with modifications introduced by the pathology team (IUSA, ULPGC) [19,20]. For the anatomical extraction of the pituitary gland, however, a novel skull-opening methodology was specifically developed for the present study. This modified protocol was applied to 22 of the necropsied dolphins (*n* = 22) across all studied species (Appendix A). For the remaining animals (*n* = 25), the skull opening was performed as described by Sacchini [21].

Fresh pituitary glands were collected from 10 common bottlenose dolphins (*Tursiops truncatus*), including five animals maintained at Loro Parque and five stranded, free-ranging individuals; 13 stranded common dolphins (*Delphinus delphis*); and 24 stranded Atlantic spotted dolphins (*Stenella frontalis*). All individuals were classified as either very fresh or fresh at the time of necropsy. The glands were analyzed macroscopically and microscopically using optical microscopy.

For the immunohistochemical study, a selection of these glands from both male and female animals—adults, juveniles, or calves—from all three studied species were included.

Samples from the pituitary glands of three of the included common dolphins (one adult female, one adult male, and one juvenile male) were preserved and processed for ultrastructural studies through electron microscopy.

Moreover, one juvenile female common dolphin that died shortly after live stranding in Gran Canaria on 22 May 2023, additionally underwent noninvasive magnetic resonance imaging (MRI) before necropsy and tissue sampling to determine the precise anatomical location and structural details of the pituitary gland (Appendix A).

All necessary permits were obtained from the Spanish Ministry for the Ecological Transition and Demographic Challenge and the Canarian Government’s Environmental Department (project number PID2021-127687NB-10). A full copy of the authorization is included in the Appendix A for reference (Appendix A).

### 2.2. Anatomical Identification of the Pituitary Gland Using Magnetic Resonance Imaging (MRI)

To gain a better understanding of the anatomical disposition of the pituitary gland in cetaceans, MRI was conducted on a subadult male common dolphin.

The MRI was performed at the Veterinary Hospital IVC Evidensia HV Tarahales (Las Palmas, Spain) within 24 hpm. During this time, the animal was kept refrigerated. A 1.5 Tesla Toshiba Vantage Elan system (Canon Medical Systems, Otawara, Japan) was employed, with the dolphin positioned in the prone orientation. A standard MRI protocol was applied to acquire spin-echo (SE) T1-weighted and gradient-echo short tau inversion recovery (GE-STIR) T2-weighted images in sagittal, transverse, and dorsal anatomical planes of the dolphin’s head [22].

### 2.3. Pituitary Gland Extraction and Sampling

The opening of the skull was performed using a swinging saw, following either the protocol established by Sacchini (*n* = 25) [21], or the modification proposed in this study (*n* = 22).

This modification extended the ventral access window to the brain, allowing for the complete extraction of the pituitary gland and infundibulum, with the gland still attached to the brain. By accessing the ventral portion of the central nervous system, this approach enabled a thorough dissection of the fibrous pad in which the pituitary gland is embedded. We first performed three incision lines in the skull as follows (Figure 1):Dorsal line: a line parallel to the nuchal ridge (crista occipitalis externa), 1 cm caudal to it;Lateral lines: two perpendicular lines extending from the dorsal line through the parietal and squamosal bones in the temporal fossa. These bilateral lines continued to the outermost lateral angles of the occipital condyles;Ventral continuation: with the head turned upside down (ventral view), the lateral lines were extended, forming a V-shaped pattern. These lines started at the lateral angles of the occipital condyles and converged just caudal to the basioccipital crest.

After making these incisions, the meninges around the medulla in the foramen magnum were dissected. The bone cap was then carefully removed, and the dural pad within the depression of the sphenoid bone was meticulously dissected using a scalpel. This step ensured that the pituitary gland, embedded within the fibrous pad, was detached while remaining connected to the brain.

After fixation by immersion in 10% neutral-buffered formalin (VWR Chemicals), both the brain and pituitary gland were sampled for histological processing. Sagittal sections of the gland were preferred, as they provide superior anatomical evaluation by enabling the simultaneous assessment of the adenohypophysis, neurohypophysis, and infundibulum.

### 2.4. Histochemical Study

For histological examination, the entire pituitary gland was embedded in paraffin and sectioned at a thickness of 5 μm. Pituitaries were stained with hematoxylin and eosin (H-E); periodic acid Schiff-Orange G (PAS-OG), Masson’s Trichrome (MT), and Masson’s Trichrome-Orange G (MT-OG) (reagents were sourced from Panreac AppliChem or VWR Chemicals) [25,26]. Imaging was performed using an Olympus BX51 microscope equipped with a DP21 camera and a 0.5X (U-TV0.5XC-3) adapter (Olympus Corp., Tokyo, Japan).

### 2.5. Immunohistochemical Study

Formalin-Fixed Paraffin-Embedded (FFPE) pituitary tissue sections (3 μm thick) were mounted on slides precoated with Vectabond (Vector Laboratories, Newark, CA, USA) were immunolabelled with polyclonal anti-ACTH Antibody (206A-74) at 1/75 dilution (Merck, Buenos Aires, Argentina); polyclonal anti-alpha-MSH Antibody (M0939-.2ML) at 1/500 dilution (Merck); and polyclonal anti-TSH Antibody (211A-14) at 1/100 dilution (Merck). Standardization was performed according to reference guides [27]. Positive control tissue consisted of a chimpanzee’s hypophysis. Negative controls included sections where the primary antibody was omitted and replaced with nonimmune homologous 10% serum in phosphate-buffered saline (PBS), as well as tissues lacking the target antigen (Appendix A).

Standard immunohistochemistry was carried out using 3-3′ diaminobenzidine (DAB) as the chromogen. Samples were deparaffinated with xylene and rehydrated through graded ethanol. Endogenous peroxidase was blocked by incubation with 3% hydrogen peroxide in methanol for 30 min at room temperature. Antigen retrieval was performed either by heat-induced epitope retrieval using citrate buffer (pH 6.0) at 96 °C for 15 min or by enzymatic digestion with 0.1% protease from Streptomyces (P5147) (Merck Laboratories) in PBS. Sections were incubated overnight at +4 °C with the primary antibodies, followed by a 30 min incubation at room temperature with a biotinylated secondary antibody. Detection was performed using the VECTASTAIN Elite ABC-Peroxidase Kit (PK-6100) (Vector Laboratories), with DAB incubation for 2–7 min. Counterstaining was performed with Mayer’s Hematoxylin for 5–10 min. Finally, the slides were mounted using Faramount Aqueous Medium (S3025) (Agilent, Beijing, China).

### 2.6. Ultrastructural Study

For the ultrastructural analysis, rectangular tissue samples measuring approximately 2 × 5 mm and less than 1 mm in thickness were obtained from one of the sagittal planes of the adenohypophysis using a sharp blade. No samples of the neurohypophysis were included in this study, as the focus was restricted to the endocrine characterization.

The samples were rinsed in phosphate-buffered saline (PBS) for 10 min, then fixed overnight at 4 °C in a 2.5% glutaraldehyde solution prepared in 0.1 M phosphate buffer (pH 7.4). After initial fixation, the samples were post-fixed in 1% osmium tetroxide (Merck Laboratories) in 0.1 M phosphate buffer (pH 7.4) for 30 min. Following dehydration through a graded ethanol series, the tissues were embedded in Araldite resin M (10951) (Merck Laboratories). Semi-thin sections were prepared using an LKB ultramicrotome and stained with toluidine blue, while ultra-thin sections were stained with uranyl acetate and lead citrate. Ultrastructural examination and imaging were performed using a JEM 1400 transmission electron microscope (TEM), (JEOL Ltd., Tokyo, Japan) at the Central Microscopy Research Facilities, Universidad de Córdoba (Spain).

Morphometric analyses were conducted using ImageJ (https://imagej.net/ij/, accessed on 2 March 2024) (National Institutes of Health, Bethesda, MD, USA). Specific ultrastructural components were measured, and the data were exported to Excel for statistical analysis and graph generation.

## 3. Results

### 3.1. Magnetic Resonance Imaging

The pituitary gland appeared on MRI as a small, compact, and well-defined oval-shaped structure. It was situated inferior to the hypothalamus and optic chiasm, within the midline region at the base of the brain on the sphenoid bone. Superiorly, it was bordered by the optic chiasm, while laterally it was adjacent to the cavernous sinuses, though the sinuses were not always clearly visualized in every section due to image resolution limitations. The floor of the gland was demarcated by the dense signal void of the sphenoid bone, while its superior connection to the hypothalamus via the infundibulum was less distinctly visible, likely due to the resolution and slice thickness of the image.

On the coronal T2-weighted MRI image, the gland exhibited intermediate signal intensity, contrasting with the high-signal cerebrospinal fluid (CSF) surrounding the skull base, which facilitated clear delineation of the gland (Figure 2).

T2-weighted imaging was particularly effective in highlighting fluid-filled spaces, such as the surrounding CSF, aiding in the differentiation of the hypophysis from adjacent structures. In contrast, T1-weighted imaging enhanced the internal anatomical contrast of the gland, providing greater structural detail against the darker CSF background.

### 3.2. Gross Anatomical Description

The modified dissection protocol developed in this study allowed for the complete, intact removal of the pituitary gland, preserving its anatomical position and connection to the brain (*n* = 22). In contrast, pituitary glands obtained following brain removal by dissecting the fibrous pad on the basisphenoid bone (*n* = 25) were all severed at the infundibulum or *pars tuberalis*, with some samples lacking the neurohypophysis.

The proposed approach ensured that both the adenohypophysis and neurohypophysis were retrieved intact, without structural damage or separation, and remained attached to the brain via the hypothalamus through the infundibulum). This technique allowed for the full visualization of the *pars tuberalis* and infundibulum in 20 of the specimens that underwent the modified protocol (*n* = 20/22). The remaining two pituitaries were torn during skull opening due to sawing errors during necropsy.

The hypophysis was observed as a small, bean-shaped organ situated at the base of the brain, with a predominantly wider lateral than anteroposterior orientation. Gross examination revealed two distinct regions: the adenohypophysis, which comprised the larger and more prominent glandular portion, and the neurohypophysis, a smaller and more compact region (Figure 3A). These regions were clearly distinguishable by differences in size, texture, and relative position, with the neurohypophysis located posteriorly and closely associated with the hypothalamus via the slender infundibular stalk. The gland was surrounded by the dura mater, which contained an aperture for the passage of the infundibulum, connecting the hypophysis to the hypothalamus just caudal to the optic chiasm.

### 3.3. Histological Description

Histological analysis of the hypophysis revealed distinct structural differences between its two main regions: the adenohypophysis and the neurohypophysis. The modified dissection protocol was crucial in preserving the gland’s structural integrity, providing an optimal view of its histological architecture in situ. This approach maintained the relationship between the adenohypophysis and neurohypophysis, as well as their connections to surrounding vascular and neural components, particularly the link to the hypothalamic region (Figure 3B). The adenohypophysis displayed a well-organized arrangement of epithelial cells forming cords, clusters, and ribbons of hormone-producing cells, interspersed within a rich, dense sinusoidal bed, and supported by a connective stroma (Figure 4B). It was enclosed in a dense, irregular connective tissue capsule. In the *pars tuberalis*, these cells were arranged in a cord-like pattern, more closely associated with numerous blood vessels (Figure 4D).

The hormone-producing cells within the parenchyma were histochemically classified as either chromophilic or chromophobic, based on their affinity for histochemical stains. Chromophilic cells were further differentiated into acidophilic and basophilic types, based on the staining characteristics of their secretory granules. Chromophobic cells appeared pale or unstained (Figure 4B).

In contrast, the neurohypophysis displayed a fibrous, sparsely cellular composition, predominantly composed of unmyelinated axons extending from the hypothalamus, alongside specialized glial cells known as pituicytes. These axons were surrounded by a dense capillary plexus and contained spherical eosinophilic hormone storage which were also faintly stained as PAS-positive structures, consistent with Herring bodies (Figure 4C). The neurohypophysis was encased in a thick and highly vascularized layer of connective tissue, clearly delineated by MT stain (Figure 4A).

The neuroendocrine connection between the pituitary gland and the hypothalamus was clearly visible through the *pars tuberalis*, where endocrine cell clusters gradually became less densely packed as they merged with the neural tissue. This transition was characterized by thick, digitiform extensions, where endocrine and neural components seamlessly integrated (Figure 3B).

### 3.4. Inmunohistochemical Study

Immunohistochemical staining of the adenohypophysis using antibodies against ACTH, MSH, and TSH revealed distinct patterns of cytoplasmic labelling common to bottlenose dolphins, common dolphins, and Atlantic spotted dolphins (Figure 5 and Figure 6).

#### 3.4.1. ACTH Labelling

ACTH-positive cells exhibited intense cytoplasmic labelling and were the most abundant among the three hormones analyzed. Staining was prominent throughout the gland, with strong labeling in both the *pars distalis*, where the cells formed large clusters, and in the *pars tuberalis*, where they were arranged in cords aligned with the sinusoids (Figure 5). ACTH-positive cells had a large, spacious cytoplasm and a slightly polygonal shape.

#### 3.4.2. MSH Labelling

Cytoplasmic labelling for MSH was observed predominantly in cell clusters or cords. These cells were more prominent in the *pars distalis* than in the *pars tuberalis*, where they appeared more sparsely distributed (Figure 6). MSH-positive cells were scattered throughout the gland.

#### 3.4.3. TSH Labelling

TSH-positive cells were less abundant than MSH and ACTH cells. These cells displayed cytoplasmic labelling, forming small clusters of two to three cells or appearing as isolated cells (Figure 7). They were predominantly localized in the *pars tuberalis*, with fewer cells observed in the *pars distalis*. Morphologically, TSH-positive cells were small and polygonal in shape.

### 3.5. Ultrastructural Description of the Adenohypophysis Through Transmission Electron Microscopy (TEM)

In the examined adenohypophyses, six distinct types of endocrine cells were identified, each exhibiting variable morphology and arrangement (Figure 8).

Additionally, follicular cells lacking secretory granules were observed. These follicular cells formed a pseudo-skeletal structure, forming tight junctions between adjacent cells. Surrounding the adenohypophysis, cortical cells encased the organ, providing further structural support.

Structurally, the adenohypophysis resembled a parenchymatous organ, with the parenchyma composed of endocrine cells and the stroma consisting of a delicate network of reticulin fibers associated with cells, follicular cavities, and numerous fenestrated capillaries. The functional regions of the adenohypophysis appeared irregular, featuring areas that ranged from palisade-like acinar zones to mosaic-like arrangements (Figure 9).

#### 3.5.1. Corticotrophs (ACTH-Producing Cells)

Corticotrophs were easily identifiable under transmission electron microscopy due to their shape and components. They were typically arranged independently or in small groups of two or three around blood capillaries. These cells generally exhibited an oval or stellate morphology, terminating in cytoplasmic pedicels of varying thickness, depending on their shape. These pedicels connected closely with the wall of fenestrated blood capillaries, establishing a tight association between the cells and the capillaries. Corticotrophs were characterized by a large, centrally located nucleus, which was spherical to oval in shape and contained uniform chromatin with moderate electron density (Figure 10A).

Among all their cellular components, their secretory granules were the most distinctive for identification purposes. These granules were the smallest among all adenohypophyseal cell types (Appendix A) and were aligned parallel to the cytoplasmic membrane, occasionally even fusing with it (Figure 8). Measurements of ACTH granule diameters across different animal groups were 198.49 nm for juvenile individuals, 204.23 nm for adult female dolphins, and 205.96 nm for adult male dolphins.

Corticotroph granules were spherical, enveloped by a clear membrane, and contained a dense, homogeneous granular core. They were arranged in small clusters within the cytoplasm, but the majority were aligned parallel to the cytoplasmic membrane, almost adhering to it (Figure 10B,D). These granules accumulated in the pedicels attached to the capillaries, forming a single row where their membranes fuse with the cytoplasmic membrane. In some granules, degradation of the granular core was observed, indicating the process of ACTH granule secretion.

The most prominent cytoplasmic organelle observed in these cells was the rough endoplasmic reticulum (RER), composed of ribosome-studded cisternae. These structures were distributed throughout the cytoplasm or clustered in specific areas near the cytoplasmic membrane, where they were closely associated with the secretory granules (Figure 10C). The Golgi complex was also highly developed and active, playing a key role in the synthesis of secretory granules. It consisted of numerous flattened sacs, transfer vesicles, and condensation vesicles.

Morphometric studies conducted using electron microscopy revealed the following percentages of corticotroph abundance across different groups: 14.55% in the juvenile male, 16.84% in the adult female, and 17.85% in the adult male.

#### 3.5.2. Lactotrophs (LTH-Producing Cells)

Lactotroph cells were isolated and irregularly distributed throughout the adenohypophyseal parenchyma, with virtually no association among themselves. Due to their pronounced pleomorphism, they were easily identifiable.

They exhibited an oval shape with a centrally located nucleus, which ranged from spherical to ovoid, and contained an ovoid nucleolus. All organelles were highly developed, with the endoplasmic reticulum being prominent, although the Golgi complex was comparatively more pronounced (Figure 11).

The secretory granules of lactotrophs were the most pleomorphic among all adenohypophyseal cells studied (Figure 11C). In these cells, the granules were generally arranged in small clusters, positioned eccentrically and sometimes polarized. The maximum diameter recorded for these granules across different animals was: 329.38 nm in juvenile individuals, 326.96 nm in adult male dolphins, and 323.94 nm in adult female dolphins.

The rough endoplasmic reticulum was highly developed and in close association with the granules and the Golgi complex. It was composed of numerous cisternae of irregular diameters (lumens) with a distinct ribosomal coating. At the peripheral zones, transfer vesicles formed, which eventually fused with the Golgi complex.

However, in these cells, the most developed organelle was the Golgi complex, which played a key role in secretory granule formation. It consisted of flattened sacs along with highly irregular condensation vesicles (Figure 11D).

Morphometric analysis showed that these cells were relatively scarce, with an abundance of 10.74% in the juvenile dolphin, 11.53% in the adult male, and 11.06% in the adult female.

#### 3.5.3. Somatotrophs (GH-Producing Cells)

Somatotropic cells, or growth hormone-producing cells, were distributed throughout the adenohypophyseal parenchyma. They were found either isolated or grouped in arrangements resembling palisades or mosaics, and they were closely associated with fenestrated capillaries. Due to their large size and the high electron density of their secretory granules, they were easily identified and well-defined under electron microscopy (Appendix A).

They had a round, oval, or elongated shape, were medium-sized, and lacked cytoplasmic extensions. They featured a very large, oval nucleus with homogeneous chromatin and one or more nucleoli (Figure 12A). Their surface was smooth, and they were in close contact with fenestrated capillaries. These cells also exhibited a highly electron-dense hyaloplasm, which, in combination with their granule deposits, made them the densest cells in the entire adenohypophysis. Additionally, they displayed a distinct mosaic or palisade arrangement.

The secretory granules were the most characteristic feature of these cells (Figure 12B). They were the most abundant among all adenohypophyseal cells, with a large, spherical structure and a highly electron-dense granular core. Granules were primarily arranged in a large central or polarized cluster, oriented toward the membrane facing the fenestrated capillaries. However, some granules were also positioned closer to the cytoplasmic membrane, likely for secretion (Figure 10).

In these cells, the granule sizes measured in different animals, sorted by age and sex, were 540.38 nm in the juvenile animal, 430.96 nm in the adult female, and 440.61 nm in the adult male.

The most developed organelle in somatotrophs was the rough endoplasmic reticulum (RER). All these cells exhibited large deposits of RER, making the region highly basophilic. These deposits were distributed throughout the cytoplasm and consisted of parallel cisternae coated with polyribosomes. RER deposits were typically concentrated in polar cytoplasmic regions, surrounded by granules and ribosomes. Additionally, these regions were often found near follicular adenohypophyseal cells.

The Golgi complex, responsible for granule formation, was also highly developed. Its main structural components—flattened membranous sacs (cisternae), transfer vesicles, and condensation or secretory vesicles (precursors of the secretory granules)—were clearly identifiable.

From morphometric studies using TEM, the following data were obtained regarding the abundance of these cells: somatotrophs were found in 21.54% of the pituitary of the juvenile male; 18.93% in the adult male, and 18.40% in the adult female.

#### 3.5.4. Gonadotrophs (GnRH -Producing Cells)

The gonadotropic cells were distributed throughout the adenohypophyseal parenchyma, typically arranged individually and closely associated with fenestrated capillaries and follicular cells. These cells were the largest among all adenohypophyseal cell types (Appendix A). In most cases, these cells could be identified under electron microscopy due to their rough endoplasmic reticulum (RER), Golgi complex, and occasionally their secretory vesicles. These cells were ovoid to triangular in shape and did not exhibit cytoplasmic projections (Figure 8). Their nucleus was oval to spherical, containing homogeneous chromatin and one or two highly visible nucleoli, while their nuclear envelope often showed indentations (Figure 13).

They were highly abundant, presumably due to their key role in regulating the ovaries and testes. Morphometric analysis revealed a relatively constant number of gonadotrophs in adult dolphins, irrespective of sex, with lower numbers observed in juveniles. The proportions were as follows: 10.29% gonadotrophs in the juvenile male pituitary parenchyma, 12.13% in the adult male, and 12.36% in the adult female.

The granules of gonadotropic cells were spherical, homogeneous, and of medium electron density, distributed throughout the cytoplasm. Each granule contained a uniform granular core enclosed within a distinct Golgi membrane. Their average diameters were 381.86 nm in the juvenile male, 385.96 nm in the adult male, and 370.90 nm in the adult female. Among these, some larger granules with higher electron density were observed and identified as lysosomes.

The RER was highly developed, extending throughout the entire cytoplasmic volume and exhibiting a distinct morphology (Figure 13D). Its cisternae were independent and dilated, forming ribosome-coated reticular sacs. Consequently, the lumen appeared filled with a uniform, low-electron-density proteinaceous material. These sacs were pleomorphic, exhibiting notable size variations. The RER maintained a close relationship with the Golgi complex and secretory granules.

Among all organelles, the Golgi complex was the most developed and active in gonadotropic cells across all examined adenohypophyses. It consisted of numerous flattened sacs with dilated lumens, shared many transfer vesicles with the RER, and exhibited a high number of condensation vesicles responsible for generating secretory granules (Figure 13C).

#### 3.5.5. Thyrotrophs (TSH-Producing Cells)

These cells were irregularly distributed in small quantities throughout the parenchyma, usually arranged individually or in groups of two to three, always positioned near blood capillaries.

Ultrastructurally, they were similar to gonadotrophs, making their identification challenging. However, they were significantly smaller, being the smallest among the hormone-producing cells (Appendix A). Morphometric analysis revealed that these cells were present in low numbers across all studied animal subgroups, with the following proportions in the adenohypophyseal parenchyma: 10.15% in the juvenile dolphin, 10.90% in the adult male, and 10.63% in the adult female.

Among their characteristics, the most notable features were their RER and oval nuclei, which occupied a large portion of the cytoplasm (Figure 14A). The nucleus contained uniform chromatin and medium electron density, typically with one to three clearly defined nucleoli and a small, distinct nucleolar organizer region (NOR).

Thyrotroph granules were relatively small and among the least developed organelles within these cells (Figure 14D). They possessed a homogeneous electron-dense core and a well-defined granular membrane closely surrounding it. These granules were concentrated around their formation zone, with very few observed near the cell membrane. A distinguishing characteristic was their association with numerous active, laminar mitochondria. The average granule size measurements were 310.16 nm in the juvenile male, 307.36 nm in the adult male, and 304.23 nm in the adult female.

The mitochondria in thyrotrophs were highly abundant, metabolically active, and evenly distributed throughout the cytoplasm. They were typically elongated but occasionally appeared ovoid, with numerous lamellae filling the cytoplasmic matrix (Figure 14C).

The RER consisted of classic parallel cisternae but also exhibited dilated, ribosome-lined sacs. These sacs contained a homogeneous, low-electron-density proteinaceous material. The Golgi complex was barely discernible; however, in some cell groups, an exceptionally well-developed Golgi complex was observed, closely associated with the RER and secretory granules.

#### 3.5.6. Melanotrophs (MSH-Producing Cells)

Melanotrophs were observed in a mosaic arrangement, organized into cellular cords, or dispersed throughout the gland in association with other cell types. The mosaic-like arrangement was more frequently found in the dorsal region of the gland, where the cells were in contact with the dura mater and capsular cells.

Morphometric analysis revealed a consistent cell size across all studied animal subgroups (Appendix A). The maximum granule size was 19.01 nm in the male juvenile dolphin, 20.22 nm in the adult female, and 19.56 nm in the adult male.

These cells were characterized by an abundance of secretory granules occupying the entire cytoplasm (Appendix A). The granules were large and spherical, with scalloped edges due to their distinct and irregular granular membrane. The granular core exhibited low electron density, and in many cases, granules lacked a defined granular body.

Melanotrophs contained numerous mitochondria distributed throughout the cytoplasm. Some exhibited lamellar structures, while others displayed large dilations of the mitochondrial cristae, making them highly pleomorphic. These cells also contained numerous ribosomes and sparse RER cisternae but exhibited significant development of the Golgi complex.

In addition to their secretory granules, MSH-producing cells contained large, isolated lysosomes with high electron density. The presence of autophagosomes, frequently appearing as myelin figures, was a common ultrastructural feature (Figure 15B).

#### 3.5.7. Follicular Cells

The studied adenohypophysis contained a population of active cells within its parenchyma that lacked secretory granules but formed follicular cavities. These cells maintained close intercellular contact through desmosomal junctions, which linked them together and facilitated the formation of cavities lined with villi. They were arranged in a reticular network that interconnected with all other endocrine cells, ultimately forming closed follicular cavities—structures from which these cells derive their name.

Morphometric analysis revealed that these cells were relatively scarce. They accounted for 10.30% of the total parenchyma in the juvenile male dolphin, 10.57% in the adult female, and 10.53% in the adult male.

These cells contained a centrally located spherical nucleus with abundant chromatin and well-developed nucleoli. They also exhibited numerous ribosomes, sparse RER cisternae, and a highly developed Golgi complex (Figure 16).

They contained abundant elongated mitochondria with lamellar cristae and occasionally other structures such as centrioles and lipid droplets. Lysosomes were also frequently observed.

#### 3.5.8. Capsular Cells

A subset of non-granular cells that resembled follicular cells but exhibited distinct ultrastructural characteristics were designated as capsular cells. These cells were externally located, forming a single-row layer surrounding the pituitary gland, with desmosomal junctions connecting them. Their morphology was similar to follicular cells, and their projections extended toward Rathke’s pouch. They were small, spherical to oval-shaped cells with irregular central nuclei (Figure 16).

Morphometric analysis determined their average size as 15.66 μm in juvenile male and female dolphins, 14.92 μm in adult males, and 14.39 μm in adult females.

These cells also lacked secretory granules, though liposomes were frequently observed. They were positioned around the parenchyma but formed invaginations that eventually interacted with follicular cells.

## 4. Discussion

Understanding species-specific morphological and physiological traits is essential for advancing comparative anatomy, animal health, and welfare. However, defining concepts such as health and welfare, particularly in wildlife, remains a significant challenge for biologists and veterinarians [28,29]. This challenge is particularly pronounced in cetaceans, where health and welfare are critical areas of research in conservation and veterinary science [30,31].

Since the early 20th century, scientific publications on stranded cetaceans have increased exponentially, primarily investigating the causes of stranding and mortality, whether natural or related to anthropogenic disturbances [19,20,32]. While all organ systems contribute to health and welfare, the neuroendocrine system is especially significant due to its central role in regulating key physiological processes related to these parameters [33,34]. The pituitary gland, a key component of the neuroendocrine system and the hypothalamic–pituitary–adrenal axis, is integral to adaptive responses to environmental and physiological challenges [35,36,37].

Although pituitary gland extraction was documented during the whaling era, most cetacean morphological studies have focused on other anatomical aspects, leaving investigations into its structure and histology relatively scarce [38,39,40]. Examining its anatomy and ultrastructure provides valuable insights with both theoretical and practical applications in endocrinology, diagnostics, and conservation.

This study provides a comprehensive morphological, histological, and ultrastructural evaluation of the pituitary gland in three dolphin species, introducing novel methodologies to enhance our understanding of this critical endocrine organ. Using immunolabeling techniques, we identified specific endocrine cells, including the first reported immunolabeling of thyrotropin-secreting (TSH) cells in the pituitary adenohypophysis of these cetaceans. Additionally, through transmission electron microscopy (TEM), we characterized distinct adenohypophyseal cell types and documented unique ultrastructural features, such as secretion granules and organelles.

### 4.1. Methodological Advances in Dolphin Pituitary Extraction

The modified skull-opening protocol and hypophyseal extraction technique developed in this study represent a pivotal advancement in cetacean anatomical research, particularly for neuroendocrine cytohistological and immunocytochemical studies. This approach ensured the full preservation of the gland and its adjacent structures, allowing for more precise analyses.

During previous studies, the gland was dissected after brain removal, from the base of the skull in the “*sella turcica*” [2,7]. This technique invariably severed the hypothalamic neuroendocrine connection of the gland and often resulted in the separation of the pituitary lobes, frequently leading to the loss of the neurohypophysis. For all the individuals assessed with this methodology (*n* = 25), the pituitary was sectioned and in many animals the neurohypophysis was lost.

The structural and spatial preservation of the gland achieved with our newly proposed methodology was crucial for subsequent histological and ultrastructural evaluations, enabling the differentiation of fine anatomical features and ensuring reliable results, with the gland’s architecture remaining intact in 90% of cases. Nevertheless, delayed necropsy or advanced carcass decomposition may compromise gland integrity, potentially limiting the effectiveness of this and other techniques.

MRI technology, performed prior to necropsy and skull opening, was instrumental in precisely locating the gland in relation to other brain structures, facilitating accurate dissection. Advances in imaging techniques are transforming anatomical and pathological research in both human and veterinary medicine [41]. However, the imaging technique also revealed its limitations: the infundibulum and *pars tuberalis* were not visualized, and the resolution was insufficient to differentiate between the adenohypophysis and neurohypophysis. These observations highlight the utility of MRI for anatomical guidance while emphasizing the need for higher-resolution imaging modalities or complementary techniques for detailed pathological assessments in future studies.

### 4.2. Morphological and Inmunohistochemical Insights

Our morphological studies revealed the intricate yet delicate connection between the pituitary gland and the hypothalamus in dolphins, specifically through the infundibulum and *pars tuberalis* in all three species studied. By preserving the integrity of both pituitary lobes, we ruled out the common artefact of *pars intermedia* loss during dissection. This enabled a comprehensive, intact study of the entire gland. Our histological findings on the adenohypophysis align with those of Panin [2,7,8]. Additionally, we present the first histological evidence of the neuroendocrine connection in cetaceans, paving the way for future studies in this field.

In the neurohypophysis, we consistently identified structures compatible with Herring bodies—accumulations of neurosecretory granules at the terminal ends of hypothalamic axons. These play a key role in storing and releasing hormones such as oxytocin and vasopressin [42,43]. Their presence in the studied dolphin species suggests a functional conservation of neurohypophyseal mechanisms. Future research should confirm the presence of these hormones and investigate additional neuropeptides and biomarkers, as has been done in other cetacean brain regions [21,44].

Our dissection protocol also facilitated a complete visualization of the vascular anatomy surrounding the pituitary gland. Notably, the highly vascularized fibrous pad surrounding the adenohypophysis in dolphins, with a histological rete structure, resembles the intracranial carotid rete found in terrestrial artiodactyls. This structure is known to function in thermoregulation and neuroprotection via countercurrent heat exchange [45,46]. In species lacking this rete, pituitary vascularization relies on the inferior hypophyseal arterial circle [47,48]. However, significant variations in cetacean intracranial vascular anatomy have been reported, with some species lacking a functional internal carotid artery or relying on vertebral or basilar arteries instead [23]. Further research is needed to explore the functional integration of these vascular structures and their species-specific adaptations, offering insights into the evolutionary modifications of cetacean neuroendocrine systems.

Through immunohistochemistry, we identified thyrotrophs (TSH-secreting cells) in cetaceans for the first time. The labelling of these cells, which play a critical role in the regulation of thyroid function via the hypothalamic–pituitary–thyroid (HPT) axis, addresses a key gap in cetacean pituitary morphology research. Thyrotrophs secrete thyroid-stimulating hormone (TSH), which acts on the thyroid gland to regulate the synthesis and release of thyroid hormones (T3 and T4), essential for metabolism, thermoregulation, and growth [49]. Given the critical role of thyroid hormones in energy balance and environmental adaptation, further investigation into thyrotroph function in cetaceans is particularly relevant. These hormones influence thermogenesis and diving physiology and could serve as bioindicators of metabolic health [50,51]. Additionally, thyroid dysfunction due to environmental stressors has been implicated in metabolic disorders and reproductive disruptions across wildlife species [52,53,54]. Investigating endocrine-disrupting pollutants, such as persistent organic pollutants (POPs), which have been linked to thyroid hormone imbalances in marine mammals [55], underscores the importance of establishing a morphological and functional framework for thyrotrophs in cetaceans.

Additionally, ACTH-secreting cells (corticotrophs) were also detected consistent with Cowan’s observations [2]; the widespread presence of corticotrophs across all studied animals underscores the need for further research into the neuroendocrine mechanisms of the stress-response system in dolphins. These cells in the adenohypophysis, are stimulated by the release of corticotropin-releasing hormone (CRH) from the hypothalamus in response to internal or external stressors, and consequently these cells secrete ACTH that then acts on the adrenal cortex, promoting the synthesis and systemic release of glucocorticoids, primarily cortisol, which modulates metabolic, immune, or behavioral responses to stress [55].

As animal welfare gains prominence in wildlife conservation—particularly for flagship species like cetaceans, where ethical concerns surrounding captivity remain a subject of debate [56,57]— there is an increasing focus on assessing stress, both at an individual and population level. Many tools and endocrine indicators are being developed and adapted from other veterinary fields, like the farming industry [58,59]. However, despite increasing efforts to evaluate stress in wildlife, a fundamental gap remains in understanding the mechanistic regulation of the hypothalamic–pituitary–adrenal (HPA) axis in cetaceans. This includes species-specific physiological adaptations and responses to various stimuli. Given the HPA axis’s central role in stress adaptation, foundational studies are essential to establish a comparative framework for neuroendocrine function in cetaceans.

Finally, the detection of MSH-secreting cells scattered throughout the adenohypophysis, rather than confined to a distinct *pars intermedia*, further challenges the hypothesis of a distinct *pars intermedia* in cetaceans [2]. This finding aligns with observations in other cetacean species, where the *pars intermedia* appears reduced or absent, potentially reflecting evolutionary adaptations similar to those seen in birds and other mammals [8,9,60]. In mammals, the *pars intermedia* is typically the primary site of melanocyte-stimulating hormone (MSH) production, playing a key role in pigmentation, energy homeostasis, and the regulation of appetite [61]. However, in species such as manatees and adult humans—both of which, like cetaceans, lack body hair and have reduced integumentary melanocytes—the *pars intermedia* is vestigial, highly reduced, or absent [62,63].

Beyond pigmentation, MSH plays a role in thermoregulation and energy balance, crucial for marine mammals adapting to variable environmental conditions. In other taxa, MSH interacts with melanocortin receptors involved in appetite control and lipid metabolism, suggesting broader endocrine functions in cetaceans [61,64]. This raises important questions about melanocortin regulation in cetaceans, particularly in relation to metabolic and endocrine adaptations to the aquatic environment.

### 4.3. Ultrastructural Study

The present ultrastructural analysis represents a significant advancement in understanding the dolphin adenohypophysis. Building on foundational studies on cetaceans [16,17], we identified and comprehensively characterized, for the first time in common dolphins, the six granular cell types of the adenohypophysis—ACTH-, LTH-, STH-, GH-, MSH-, and TSH-secreting cells—as well as two non-granular cell types, follicular and capsular cells. The granular cell types were primarily distinguished by their secretory granules, complemented by ultrastructural features such as organelles and cellular morphology, which also facilitated the identification of the non-granular follicular and capsular cells. This study provides a valuable morphometric reference for future ultrastructural analyses in these and other cetacean species.

Corticotrophs, responsible for secreting adrenocorticotropic hormone (ACTH), were characterized by highly electron-dense granules, predominantly distributed towards the periphery of the cytoplasm, and the smallest granules among all endocrine cells (Appendix A). This distribution likely reflects the high turnover and secretion rate typical of these cells, consistent with findings in other mammals [65,66].

Lactotrophs exhibited larger, polymorphic granules with peripheral electron lucency and a well-developed Golgi complex. These cells were more abundant in adult females, consistent with their role in lactation and reproductive physiology. The pleomorphic nature of lactotroph granules was a distinctive feature, also observed in other species, emphasizing their complex functional role in hormone synthesis and secretion [67,68].

Somatotrophs, which are responsible for secreting growth hormone, were identified by their large size, densely packed secretory granules, and prominent RER. These characteristics reflect their active secretory function and align with findings in terrestrial mammals such as rats, bats, and carnivores like foxes [69,70,71]. The abundance and size of somatotrophs in juvenile individuals further underscore their pivotal role in supporting growth during early life stages.

Gonadotrophs, responsible for secreting follicle-stimulating hormone (FSH) and luteinizing hormone (LH), could not be ultrastructurally differentiated due to limitations in image quality and tissue preservation. These cells were equally present in male and female dolphins, though they were less abundant in juveniles, similar to findings in other species and reflecting their role in reproductive physiology [72].

In line with our immunohistochemical results, a distinct *pars intermedia* composed solely of melanotrophs was not identified. Melanocyte-stimulating hormone (MSH)-producing cells were arranged in a unique mosaic-like pattern near the periphery of the gland, with additional scattered MSH-producing cells observed throughout the adenohypophysis. These findings corroborate previous immunohistochemical observations, which described widespread α-MSH labelling in the parenchyma [8].

In this study, we identified thyrotrophs in cetaceans for the first time. These cells were characterized by their abundant mitochondria and relatively sparse granules, distinguishing them from other adenohypophyseal cell types. Their identification addresses a gap in understanding metabolic and thyroid-related morphology and physiology in marine mammals.

Follicular (stellate) cells, first described ultrastructurally in cetaceans by Young [16], were further characterized in this study. Lacking secretory granules, these cells demonstrated features indicative of intrapituitary communication, such as gap junctions and vesicle trafficking. Moreover, through their interconnected network, they may serve as a support for the endocrine parenchyma by forming a complex three-dimensional network. We also examined follicular cavities, which serve as sites for the storage and degradation of non-secreted hormones or degenerated cells [73]. Their close association with perivascular spaces suggests a role in facilitating rapid hormone release into the bloodstream, a feature briefly noted but not fully explored in earlier studies [74,75].

Finally, we also identified capsular cells for the first time in cetacean species, although they have been described in other animals such as fish and mice [76,77]. These cells, a type of nongranular adenohypophyseal cell, resemble follicular cells but display distinct ultrastructural characteristics. They formed a single row lining the gland, surrounding the adenohypophyseal parenchyma and creating a capsular structure. Capsular cells were connected by desmosomal junctions, presented metabolic fat in their cytoplasm, and exhibited epithelial-like morphology, though they lacked microvilli. While their precise function remains unclear, these cells may play a structural or organizational role in maintaining the integrity of the pituitary gland, as suggested by findings in other species [78].

Recent studies have highlighted the presence of stem cell populations in the hypophysis that contribute to pituitary homeostasis and respond to physiological demands or injuries [78]. These cells are also placed lining the margin of the adenohypophyseal lobe [79]. The structural arrangement and characteristics of capsular cells raise the possibility that they represent a population of pituitary stem cells. Further research is needed to investigate this potential, particularly through the use of immunohistochemical techniques to assess the expression of known stem cell markers.

Certain limitations were present in our study, notably the small sample size for electron microscopy, which may constrain the generalizability of our findings. Limited sample sizes may preclude our ability to detect individual or interspecies variations and to assess different physiological states (e.g., seasonal changes, reproductive states, etc.), all of which could influence adenohypophyseal ultrastructure [80,81,82]. Additionally, our study was restricted to only three species of delphinids, limiting our capacity to extrapolate these findings to the entire cetacean clade. Given the morphological and physiological diversity across cetaceans, further studies encompassing a broader range of taxa are necessary to fully characterize the variability in endocrine cell structure and function within this group.

Despite these limitations, our research holds significant potential applications in morphology and physiology, pathology, and also conservation of cetaceans. Neuroendocrine indicators may serve as biomarkers for different stressors, such as environmental changes, disease states, or reproductive health [83,84,85,86]. Given that the pituitary gland is one of the least studied endocrine organs in cetaceans, this research is of paramount importance across multiple fields. Expanding future studies to include a broader range of individuals, species, and physiological states, as well as further research into both the adenohypophysis and neurohypophysis, could provide a more holistic understanding of pituitary dynamics. Additionally, employing advanced imaging techniques and interdisciplinary approaches, including molecular biology or proteomics, would further elucidate the regulatory mechanisms of cetacean pituitary function, paving the way for non-invasive diagnostic tools and welfare efforts.

## 5. Conclusions

This study provides the first comprehensive characterization of the dolphin pituitary gland through a multidisciplinary approach, establishing a foundational understanding of its anatomy, morphology, and ultrastructure. By integrating non-invasive imaging techniques, a novel necropsy protocol, and detailed ultrastructural analyses, we have identified key structural features in three different dolphin species. Notably, the first ultrastructural identification of six granular and two non-granular adenohypophyseal cell types, along with the immunohistochemical identification of thyrotrophs, represents a significant advancement in cetacean neuroendocrinology.

The methodological advancements, including modifications to pituitary extraction protocols and the integration of imaging with histological and ultrastructural techniques, enabled the first histological visualization of the neuroendocrine connection, paving the way for more detailed anatomical and physiological investigations in cetaceans. While challenges such as sample preservation and interspecies variability remain, this study establishes a critical foundation for future research.

Finally, these findings have broad implications for species-specific health assessments, welfare practices, and conservation strategies, especially in light of the unique physiological adaptations of dolphins. Future interdisciplinary studies incorporating advanced techniques could further elucidate the regulatory mechanisms of the pituitary gland, supporting conservation efforts and non-invasive diagnostics in marine mammal research.

## Figures and Tables

**Figure 1 animals-15-00735-f001:**
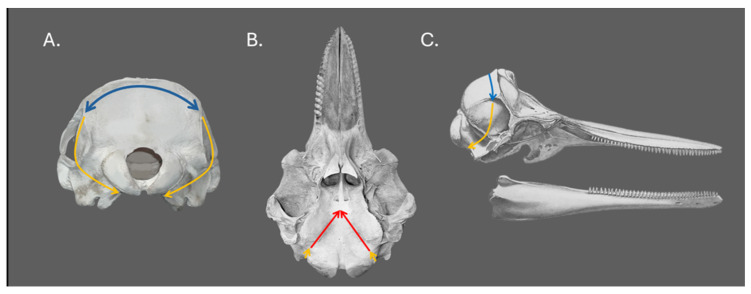
Modified skull-opening protocol for dolphins. The image shows a caudal view (**A**), ventral view (**B**), and lateral view (**C**) of the dolphin’s skulls. Dorsal line (blue), lateral lines (yellow), and ventral continuation lines (red). Images edited from Cozzi and Elsberry [23,24].

**Figure 2 animals-15-00735-f002:**
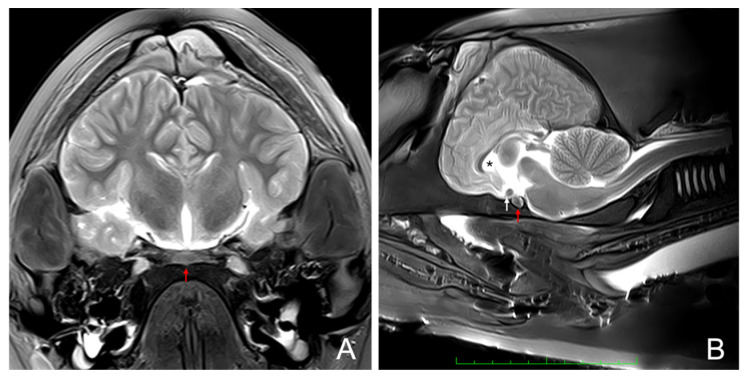
(**A**) T2-weighted MRI coronal section of a dolphin’s head showing major brain structures. The pituitary gland (red arrow) is located at the base of the brain, above the sphenoid bone and ventral to the hypothalamus. It appears as a small, well-defined structure with intermediate signal intensity, framed by the bilateral optic tracts. (**B**) T2-weighted MRI sagittal section of a dolphin’s head showing major brain structures. The pituitary gland (red arrow) is located caudal to the optic chiasm (white arrow). The cerebrospinal fluid in the ventricles (asterisk) is visible with high signal intensity.

**Figure 3 animals-15-00735-f003:**
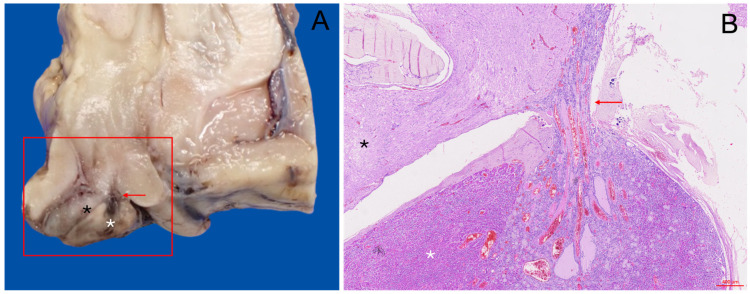
Macroscopic and microscopic sagittal views of a common dolphin’s pituitary gland. (**A**) Formalin-fixed common dolphin’s brain showing the hypothalamus, optic chiasm, and pituitary gland. The adenohypophysis (white asterisk) is located cranio-ventrally to the neurohypophysis (black asterisk). The infundibular stalk connecting the hypophysis to the hypothalamus is also apparent (red arrow). (**B**) Histological view stained with H-E at 4× magnification. Both the neurohypophysis (black asterisk) and the adenohypophysis in its *pars distalis* (white asterisk) are observed. The adenohypophysis is continued by its *pars tuberalis* at the infundibular stalk level (red arrow). The neuroendocrine connection between the endocrine tissue and nervous tissue is clearly demarcated.

**Figure 4 animals-15-00735-f004:**
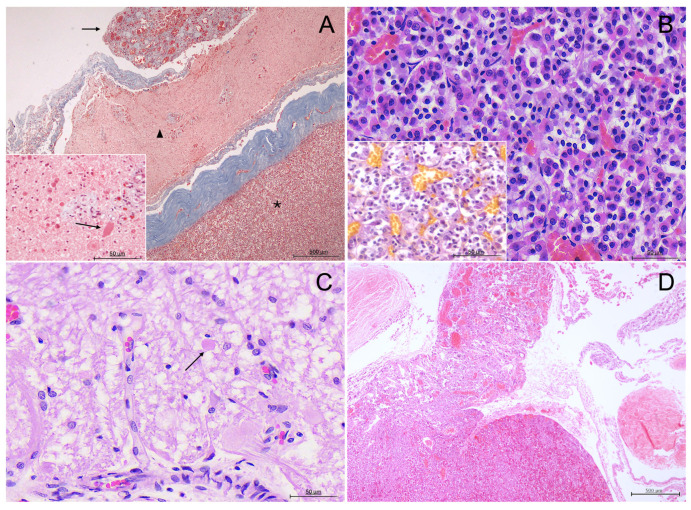
Histological and histochemical characterization of the pituitary gland in delphinid species. (**A**) Masson’s Trichrome (MT) staining at 4× magnification in a common bottlenose dolphin. The adenohypophysis (asterisk) and neurohypophysis (triangle) are distinctly separated by a thick band of dura mater and connective tissue; a highly vascular rete-like structure envelops the gland (arrow). The inset shows the neurohypophysis (MT) at 40× magnification with structures compatible with Herring bodies (arrow). (**B**) Hematoxylin-Eosin (H-E) staining at 40× magnification of the adenohypophysis of an Atlantic spotted dolphin. Acidophils, basophils, and chromophobes are arranged in cell clusters. The inset shows the same region stained with PAS-OG. (**C**) High-magnification (40×) view of the neurohypophysis in a common dolphin, stained with H-E. The image reveals unmyelinated axons, pituicytes, and structures compatible with Herring bodies (arrow). (**D**) H-E stained pituitary gland of a newborn common bottlenose dolphin at 4× magnification. The image shows the highly vascularized *pars tuberalis* and subjacent *pars distalis*.

**Figure 5 animals-15-00735-f005:**
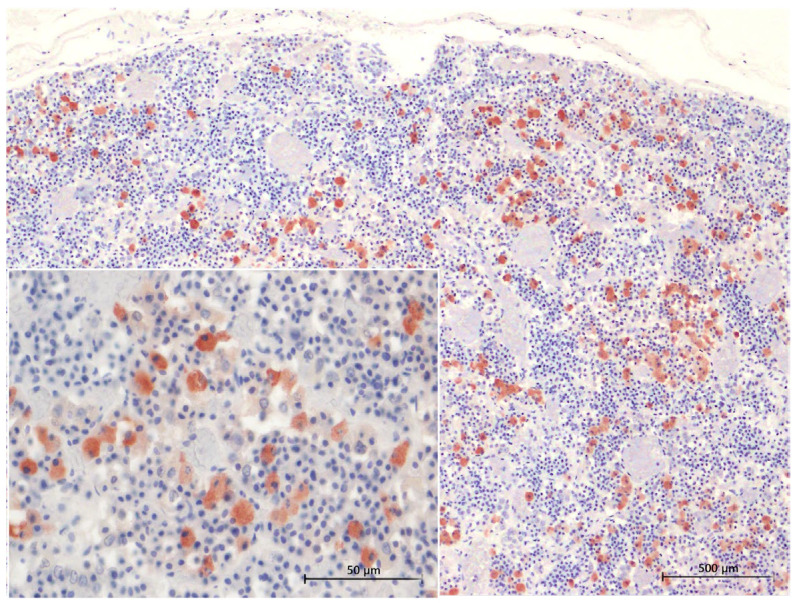
Immunohistochemical characterization of adenohypophyseal cell populations in common bottlenose dolphins using specific antibodies to anti-ACTH at 4× magnification. ACTH-positive cells exhibit strong cytoplasmic labelling and are individually dispersed or organized in small groups within the adenohypophysis. The inset provides a higher magnification (40×) view, highlighting the distinct labelling pattern in the cytoplasm of ACTH-expressing cells.

**Figure 6 animals-15-00735-f006:**
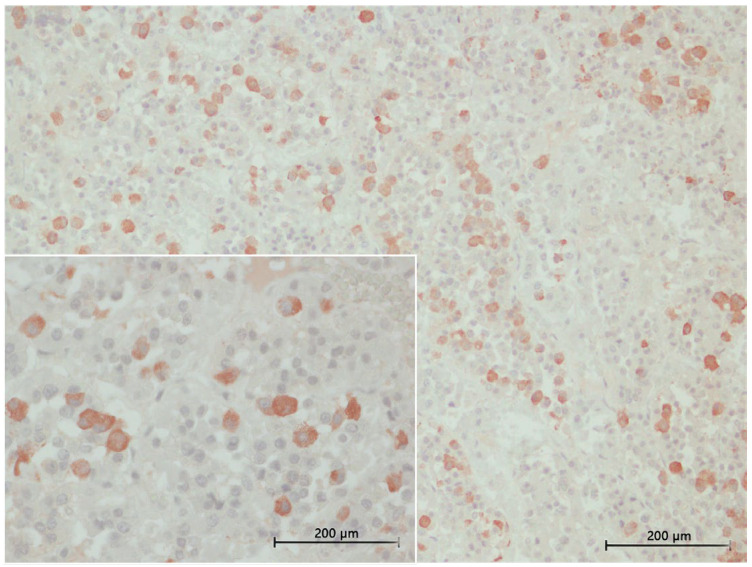
Immunohistochemical characterization of adenohypophyseal cell populations in common bottlenose dolphins using specific antibodies to anti-MSH antibody at 4× magnification. MSH-positive cells are distributed throughout the adenohypophysis, following the arrangement of endocrine cords. Cytoplasmic staining is evident. The inset shows a 40× magnification, offering a detailed view of the cytoplasmic labelling.

**Figure 7 animals-15-00735-f007:**
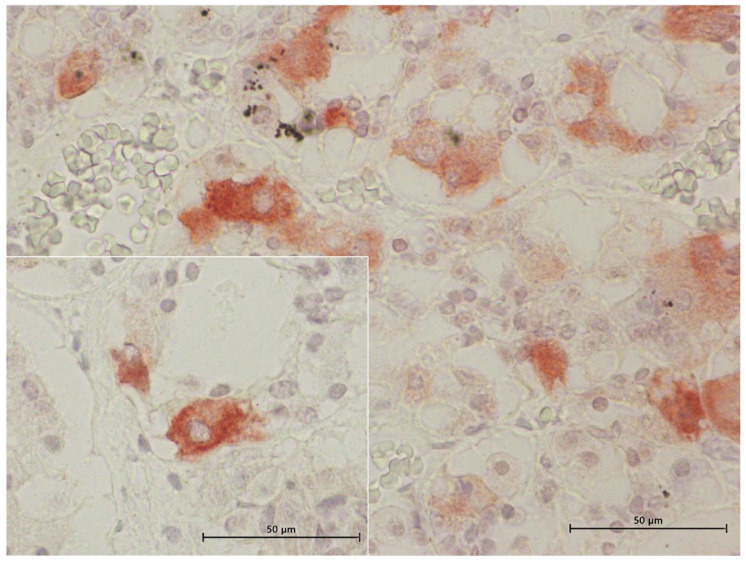
Immunohistochemical characterization of TSH-producing cells in common bottlenose dolphins. The image shows a 40× magnification, highlighting TSH-positive cells scattered individually throughout the adenohypophysis. TSH-positive cells exhibit a polygonal shape with distinct cytoplasmic staining. The inset (40× magnification) presents another region of the gland, confirming the dispersed distribution pattern of TSH-expressing cells.

**Figure 8 animals-15-00735-f008:**
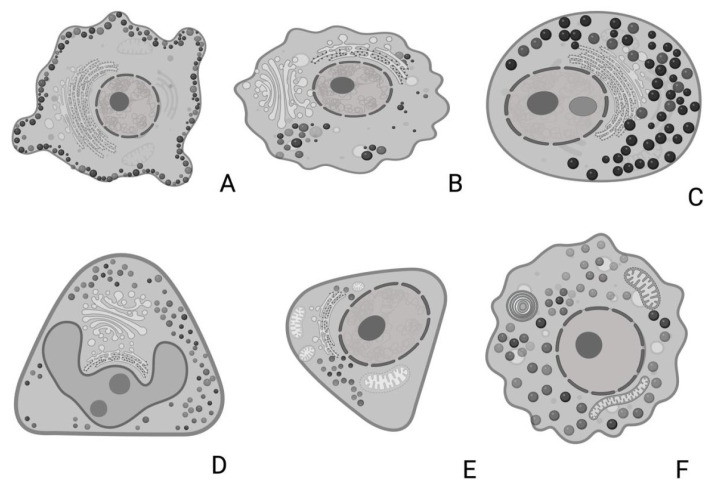
Schematic representation of the general morphology and ultrastructural characteristics (overall cellular and nuclear shape and size, secretion granules size and density, nucleus, rough endoplasmatic reticulum, Golgi complex, mitochondria) of the six different hormone-producing endocrine cells in the common dolphin (*Delphinus delphis*) pituitary gland. (**A**) Corticotrophs (**B**) Lactotrophs, (**C**) Somatotrophs, (**D**) Gonadotrophs, (**E**) Thyrotrophs, and (**F**) Melanotrophs. The illustration highlights key morphological and ultrastructural features of each cell type, as described in the text. Created in BioRender, version 04.

**Figure 9 animals-15-00735-f009:**
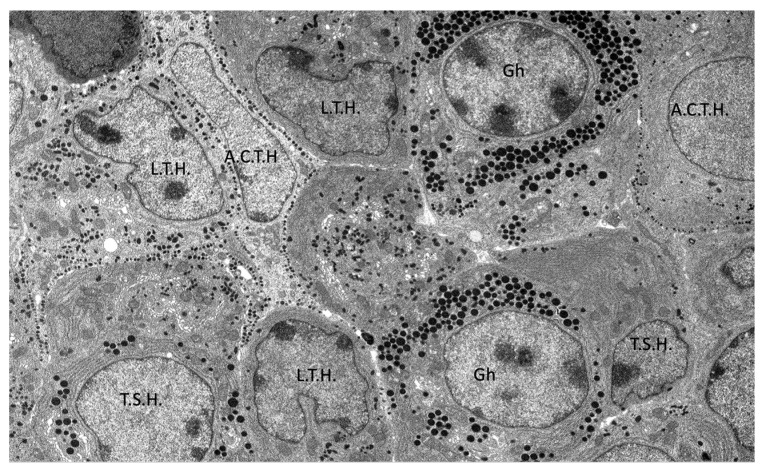
Transmission electron microscopy images of the adenohypophysis structure and identification of different hormone-producing cells in common dolphins (*Delphinus delphis*). Abbreviations: A.C.T.H., corticotrophs; L.T.H., lactotrophs, Gh, somatotrophs, T.S.H., thyrotrophs.

**Figure 10 animals-15-00735-f010:**
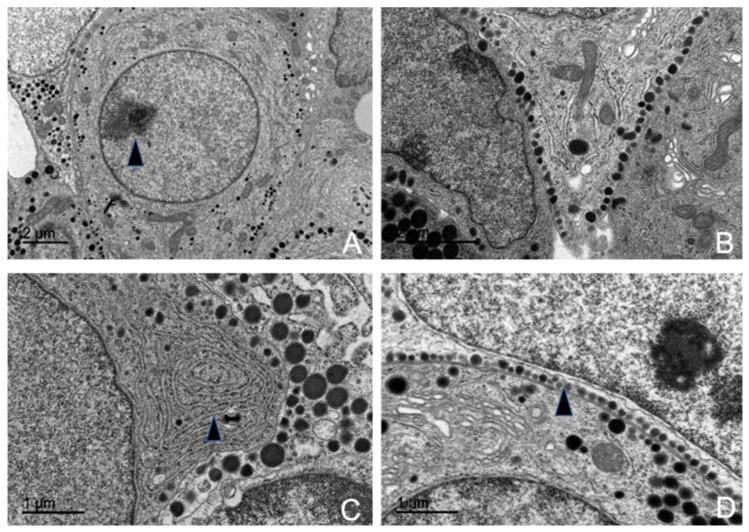
Transmission electron microscopy images of ACTH-producing cells in common dolphins *(Delphinus delphis)*. (**A**) The cell shows a large central nucleus with a visible nucleolus (arrow). (**B**) ACTH cytoplasmic projection with aligned granules in the cell membrane. (**C**) Abundant rough endoplasmic reticulum arranged in concentric layers (arrow). (**D**) Aligned ACTH electrodense granules along the cytoplasmic membrane (arrow), prepared for secretion.

**Figure 11 animals-15-00735-f011:**
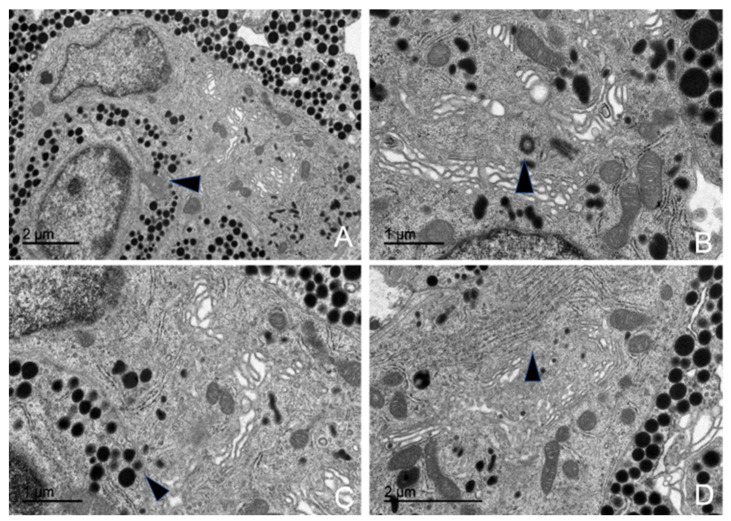
Transmission electron microscopy images of LTH-producing cells in common dolphins (Delphinus delphis). (**A**) The cell exhibits an ovoid morphology, characteristic of LTH-producing cells (arrow). (**B**) Presence of centrioles, which play a role in intracellular transport and cellular organization (arrow). (**C**) The cytoplasm contains highly pleomorphic granules, varying in size and shape, indicative of secretory activity (arrow). (**D**) A well-developed Golgi complex is observed (arrow); mitochondria are also abundant.

**Figure 12 animals-15-00735-f012:**
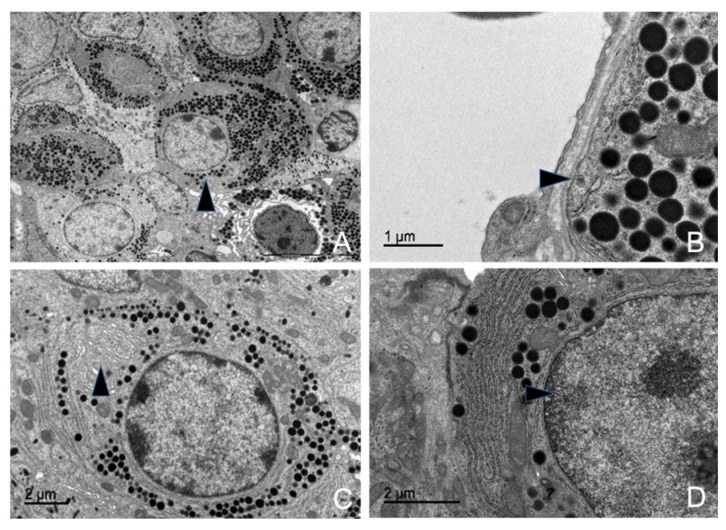
Transmission electron microscopy images of GH-producing cells in common dolphins (*Delphinus delphis*). (**A**) The cytoplasm contains abundant granules distributed throughout (arrow). The cell exhibits an ovoid morphology (arrow), a typical feature of somatotrophs. (**B**) Large, spherical, electron-dense granules are closely attached to the cytoplasmic membrane (arrow), suggesting active secretion. (**C**) A well-developed Golgi complex occupies a large cytoplasmic area (arrow), reflecting its role in processing and packaging secretory granules. (**D**) The nucleus appears spherical with a prominent nucleolus (arrow).

**Figure 13 animals-15-00735-f013:**
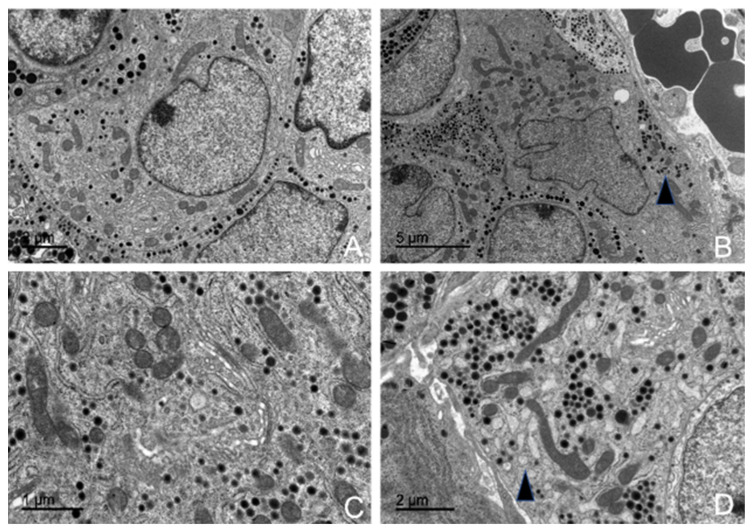
Transmission electron microscopy images of GnRH-producing (gonadotroph) cells in common dolphins (*Delphinus delphis*). (**A**) The cell exhibits an ovoid shape with a centralized nucleus. (**B**) A pleomorphic nucleus is observed (arrow), indicating nuclear variability in these endocrine cells. (**C**) A well-developed Golgi complex is prominent, reflecting its role in hormone processing and packaging. (**D**) Dilated rough endoplasmic reticulum cisternae with homogeneous content are evident (arrow), suggesting active protein synthesis. Additionally, numerous mitochondria are present throughout the cytoplasm.

**Figure 14 animals-15-00735-f014:**
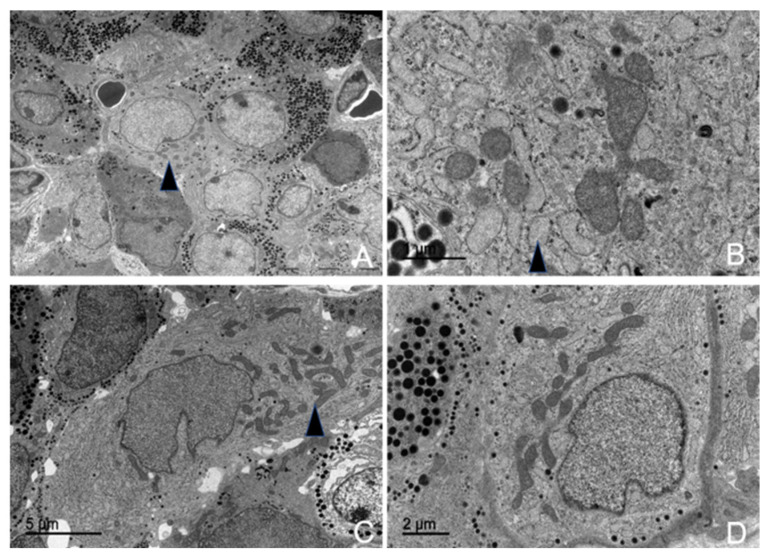
Transmission electron microscopy images of thyrotrophs in common dolphins (*Delphinus delphis*). (**A**) The cell exhibits an ovoid shape at low magnification (arrow). (**B**) Dilated rough endoplasmic reticulum cisternae are observed, appearing as sac-like structures with clear, homogeneous content (arrow). (**C**) Numerous mitochondria, particularly with a lamellar structure (arrow), are present, suggesting high metabolic activity. (**D**) Medium-sized granules are distributed within the cytoplasm.

**Figure 15 animals-15-00735-f015:**
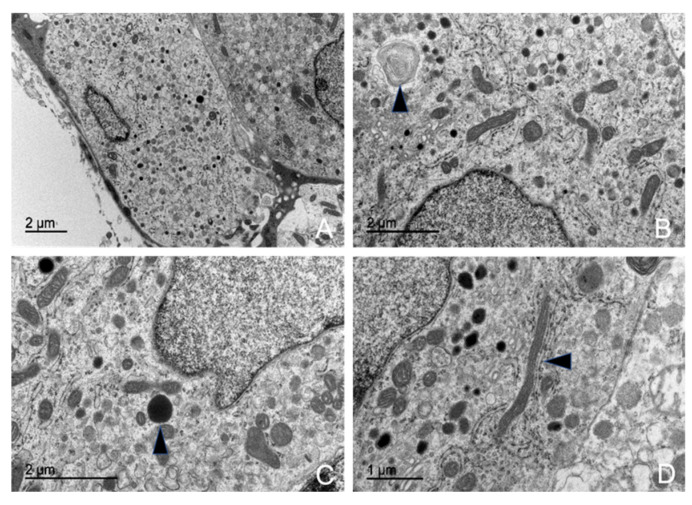
Transmission electron microscopy images of MSH-producing cells in a common dolphin (*Delphinus delphis*). (**A**) The cell exhibits an ovoid shape. Abundant granules with varying content are visible within the cytoplasm. (**B**) Myelin figures are observed (arrow). (**C**) Large lysosomes are present (arrow), likely involved in cellular digestion and turnover. (**D**) A well-developed Golgi complex with numerous flattened sacs is observed (arrow), involved in processing and packaging hormones. Additionally, mitochondria are present throughout the cytoplasm, indicating high metabolic activity.

**Figure 16 animals-15-00735-f016:**
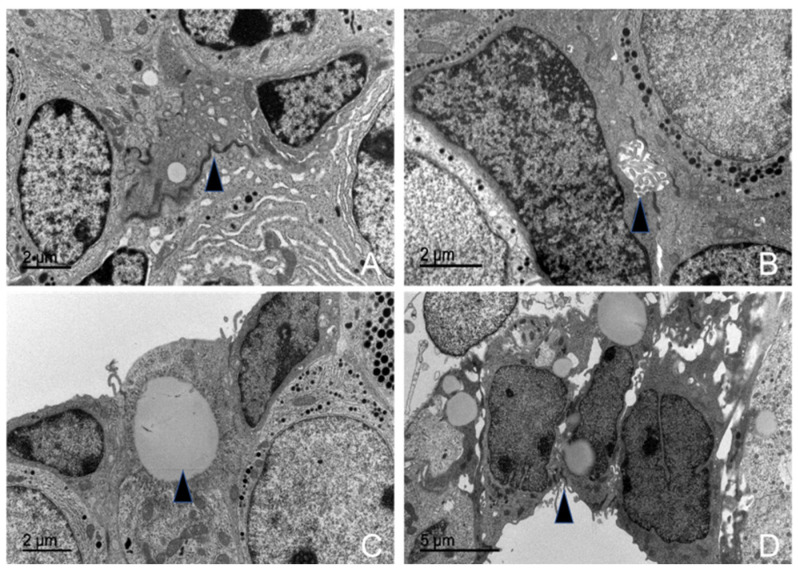
Transmission electron microscopy images of follicular and capsular cells in the adenohypophysis of common dolphins (*Delphinus delphis*). (**A**) Desmosomes are visible between the follicular cells, suggesting cell–cell adhesion (arrow). (**B)** A follicular cavity is observed, characteristic of the structure of these cells (arrow). (**C**) In capsular cells, the presence of fat droplets is evident, indicating lipid storage (arrow). Additionally, the capsular cells form a thin lining around the adenohypophysis. (**D**) The capsular cells are arranged in a pseudoepithelial palisade (arrow).

## Data Availability

The data on stranded animals and associated information are owned by the Canarian local government and are publicly available upon request. These data are held by the University of Las Palmas de Gran Canaria (ULPGC). Additional data generated and analyzed in this study may be available upon reasonable request.

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
