# Peer review of "Dolphin Pituitary Gland: Immunohistochemistry and Ultrastructural Cell Characterization Following a Novel Anatomical Dissection Protocol and Non-Invasive Imaging (MRI)"

_animals, 2025, doi:10.3390/ani15050735_

Round 1

Reviewer 1 Report

Comments and Suggestions for Authors

Reduce abstract to 200 words

Line 134: add a table/figure showing how the sampling and all the groups were managed.

Line 135: add a table where the number, age, and sex of animals for each experiment is indicated

Line 135: add the authorization number

Paragraph histochemical study: add catalogue number for all products used

Paragraph immunohistochemical study: add catalogue number for all products used

Paragraph ultrastructural study: add catalogue number for all products used

Figure 3: add scale bar

Figure 4: add scale bar in all panel

Figure 5B: move figure at line 356. Add scale bar

Figure 6: add scale bar

Add in supplementary material a panel showing all positive and negative control of immunohistochemistry

Figure 10: move figure at line 559

Figure 11: move figure at line 599

In the discussion expand the explanation of the role of ACTH, MSH and TSH

Comments on the Quality of English Language

The English could be improved, especially in the introduction

Author Response

Thank you very much for taking the time to review this manuscript. Please find the detailed responses to your insightful comments below, with the corresponding changes marked in red, as well as these corrections in the re-submitted files.

Comment 1: Reduce abstract to 200 words

Response 1: Thank you for this note, abstract has been reduced to 200 words. 

Comment 2: Line 134: add a table/figure showing how the sampling and all the groups were managed.

Response 2: Thank you for this suggestion, I agree that it adds clarification on how the animals were distributed for all the different techniques used in this study. We have added a table in supplementary materials (Table S1) that includes all studied individuals with their species, their age, sex, conservation status, the skull opening protocol utilised for each individual, as well as other complementary technique performed (IHC, TEM or MRI, besides histology). This has been accordingly referenced in the manuscript, in page 3, paragraph 3, line 116 "[(...) This modified protocol was applied to 22 of the necropsied dolphins (n = 22) across all studied species (Table S1). ]"

Comment 3: Line 135: add a table where the number, age, and sex of animals for each experiment is indicated

Response 3: We appreciate the reviewer for raising this matter. In line with the previous comment, we added a full table where all the details of the individuals were included, both from a metodological perspective, but also including the suggested data by the reviewer. This is provided as Table S1 in supplementary materials, and referenced both in Page 3, Paragraph 3 line 116 as stated above, as wel as in Page 3, Paragraph 7, Line 133.

Comment 4: Line 135: add the authorization number

Response 4: Thank you so much for noting this out. We have added the project's number in line 136 " All necessary permits were obtained from the Spanish Ministry for the Ecological Transition and Demographic Challenge and the Canarian Government’s Environmental Department (project number PID2021-127687NB-10). A full copy of the authorization is included in the Supplementary Materials for reference (Document S1)." Moreover we have provided a full copy of the authorization in the Supplementary Materials as suggested by peer review, for further clarification on authorizations.

Comment 5: Paragraph histochemical study: add catalogue number for all products used

Response 5: Thank you for noting this. We have included the laboratories our reagents are sourced from and hope this can give more clarity for our methodology, however, for the different histochemical techniques, there are several reagents standarily used and the references also vary depending on the availability over time, and given that the mentioned stains are standard histochemical stains, and not experimental or modifications, we havent specified all reactives. We have included our updates in Page 5, Paragraph 2 "histochemical study", line 184 "[Pituitaries were stained with hematoxylin and eosin (HE); periodic acid Schiff-Orange G (PAS-OG), Masson’s Trichrome (MT), and Masson’s Trichrome-Orange G (MT-OG) (reagents were sourced from Panreac Applichem or VWR Chemicals) [25,26]."] 

Moreover, here I attach a picture of the different products used as main reagents for the standard stains (hematoxylin and eosin; PAS, MT, and Orange G) used in our laboratory, for visual reference. 

Comment 6: Paragraph immunohistochemical study: add catalogue number for all products used

Response 6: Thank you for noting this as well. We have included the remaining reagents laboratories and references of the lacking products, Page 5, Paragraphs 2 and 3. For general laboratory fungibles and products, the reference was not included, as they change over time depending on the laboratories availability and budget. 

Comment 7: Paragraph ultrastructural study: add catalogue number for all products used

Response 7: Similarly to the immunohistochemical methodology, we have added the reagent's laboratories used and references when applicable, Page 5 paragraphs 4 and 5. For general laboratory fungibles and products, the reference was not included, as they change over time depending on the laboratories availability and budget. 

Comment 8: Figure 3: add scale bar

Response 8: I appreciate the reviewers comment on this matter. We have added scale bar in Page 7, between line 278 and 279).

Comment 9: Figure 4: add scale bar in all panel

Response 9: Thank you so much for noting this as well, we have modified the panel with the appropiate scale bars for each magnification. Page 8, between lines 318 and 319

Comment 10: Figure 5B: move figure at line 356. Add scale bar in Figure 3.

Response 10: We agree that moving the figure fits better with the text flow and comprehension, we have moved it to Page 9, line 342, as lines have moved with all new revisions. We have also added corresponding scale bars. 

Comment 11: Figure 6: add scale bar

Response 11: We have added scale bars as suggested, and appreciate this revision and comment. 

Comment 12: Add in supplementary material a panel showing all positive and negative control of immunohistochemistry

Response 12: This is a very useful and important point in our opinion and we have added in supplementary material a panel with the positive and negative controls for the three antibodies used in immunohistochemistry (Figure S2). We have stated this in the main text in the methodology for immunohistochemistry, page 5, paragraph 2, line 197 "[Positive control tissue consisted of a chimpanzee’s hypophysis. Negative controls included sections where the primary antibody was omitted  and replaced with nonimmune homologous 10% serum in phosphate-buffered saline (PBS), as well as tissues lacking the target antigen (Figure S2).  ]" 

Comment 13: Figure 10: move figure at line 559 

Response 13: Thank you for your comments regarding the correct positioning of the ultrastructural images. We have noted there were some mistakes on the numbering on a couple of images, so we have looked at them in detail and corrected to then properly adjust the images position according to your suggestions. We have then moved Figure 10 to line 429, since line 559 corresponds to another type of cell.

Comment 14: Figure 11: move figure at line 599

Response 14: Similarly to the previous figure, we have adjusted its position and moved it to line 462, in page 16, so its in the gonadotroph section that it refers to. 

Comment 15: In the discussion expand the explanation of the role of ACTH, MSH and TSH

Response 15: We appreciate the reviewer’s suggestion. We have expanded the discussion on the roles of ACTH, MSH, and TSH, providing additional context on their physiological functions and relevance in cetaceans. These revisions help to better integrate our findings with broader endocrine and comparative physiological perspective. We have further developed the discussion on these three types of cell in page 21 and 22, from lines 767 to 816. "[ Through immunohistochemistry, we identified thyrotrophs (TSH-secreting cells) in cetaceans for the first time. The labelling of these cells, which play a critical role in the regulation of thyroid function via the hypothalamic-pituitary-thyroid (HPT) axis, addresses a key gap in cetacean pituitary morphology research. Thyrotrophs secrete thyroid-stimulating hormone (TSH), which acts on the thyroid gland to regulate the synthesis and release of thyroid hormones (T3 and T4), essential for metabolism, thermoregulation, and growth [49]. Given the critical role of thyroid hormones in energy balance and environmental adaptation, further investigation into thyrotroph function in cetaceans is particularly relevant. These hormones influence thermogenesis and diving physiology and could serve as bioindicators of metabolic health [50,51] . Additionally, thyroid dysfunction due to environmental stressors has been implicated in metabolic disorders and reproductive disruptions across wildlife species [52–54]. Investigating endocrine-disrupting pollutants, such as persistent organic pollutants (POPs), which have been linked to thyroid hormone imbalances in marine mammals [55], underscores the importance of establishing a morphological and functional framework for thyrotrophs in cetaceans.

Additionally, ACTH-secreting cells (corticotrophs) were also detected consistent with Cowan’s observations [2]; the widespread presence of corticotrophs across all studied animals underscores the need for further research into the neuroendocrine mechanisms of the stress-response system in dolphins. These cells in the adenohypophysis, are stimulated by the release of corticotropin-releasing hormone (CRH) from the hypothalamus in response to internal or external stressors, and consequently these cells secrete ACTH that then acts on the adrenal cortex, promoting the synthesis and systemic release of glucocorticoids, primarily cortisol, which modulates metabolic, immune or behavioral responses to stress [55].

As animal welfare gains prominence in wildlife conservation—particularly for flagship species like cetaceans, where ethical concerns surrounding captivity remain a subject of debate [56,57]— there is an increasing focus on assessing stress, both at an individual and population level . Many tools and endocrine indicators are being developed and adapted from other veterinary fields, like the farming industry [58,59]. However, despite increasing efforts to evaluate stress in wildlife, a fundamental gap remains in understanding the mechanistic regulation of the hypothalamic-pituitary-adrenal (HPA) axis in cetaceans. This includes species-specific physiological adaptations and responses to various stimuli. Given the HPA axis's central role in stress adaptation, foundational studies are essential to establish a comparative framework for neuroendocrine function in cetaceans.

Finally, the detection of MSH-secreting cells scattered throughout the adenohypophysis, rather than confined to a distinct pars intermedia, further challenges the hypothesis of a distinct pars intermedia in cetaceans [2]. This finding aligns with observations in other cetacean species, where the pars intermedia appears reduced or absent, potentially reflecting evolutionary adaptations similar to those seen in birds and other mammals [8,9,60]. In mammals, the pars intermedia is typically the primary site of melanocyte-stimulating hormone (MSH) production, playing a key role in pigmentation, energy homeostasis, and the regulation of appetite [61]. However, in species such as manatees and adult humans—both of which, like cetaceans, lack body hair and have reduced integumentary melanocytes—the pars intermedia is vestigial, highly reduced, or absent. [62,63] Beyond pigmentation, MSH plays a role in thermoregulation and energy balance, crucial for marine mammals adapting to variable environmental conditions. In other taxa, MSH interacts with melanocortin receptors involved in appetite control and lipid metabolism, suggesting broader endocrine functions in cetaceans [61,64]. This raises important questions about melanocortin regulation in cetaceans, particularly in relation to metabolic and endocrine adaptations to the aquatic environment. "]

Comment 16 (English quality): The English could be improved, especially in the introduction

Response 16: We appreciate the reviewer’s comment regarding the English quality. We have carefully revised the manuscript, particularly the Introduction, to improve clarity, readability, and overall language quality. Changes have been made throughout the text to enhance precision and fluency, and can be found and tracked in the attached new version of the manuscript.

Reviewer 2 Report

Comments and Suggestions for Authors

Review of Alonso-Almorox et al.’s manuscript

This manuscript represents a well-done study of the pituitary gland in three dolphin species. The work includes a new dissection protocol, magnetic resonance analysis and transmission microscope observations to characterize the gross morphology and the ultrastructure of the pituitary gland providing new and important data. The manuscript is well-written and well-organized and includes all the necessary texts and illustrations. As a whole, I think that it may be accepted for publication pending the very minor changes that I am going to list in the following text.

First, I strongly recommend the authors to provide a line drawing of a generalized mammalian hypophysis illustrating the three main portions of this gland. Such an image would be of great help to the non-specialized reader to understand the topological relationships of the different portions of the gland otherwise the jargon terms may represent a barrier to the understanding. Then, I suggest to include in the same image another line drawing with the main characters of the dolphin’s pituitary gland showing the main portions and illuminating the main differences with respect to the generalized mammalian pituitary gland.

Second, I recommend the authors to provide line drawings illustrating the different morphological and ultrastructural characters of the cells they described in the text. There is plenty of examples of very nice scientific illustrations of endocrine cells in the literature and adding such an illustration would be of great help for the non-specialized reader as it would provide information that is immediately understandable by everyone as the differences described in the text seem evident. Unfortunately, it is hard to discern peculiar characters of the different cell types from the images obtained by the transmission microscopy and this makes it harder both the reading and understanding.

Third, I recommend the authors to emphasize that their results are from three dolphin species (Delphinidae family) and cannot be obviously generalized to the whole Cetacea clade. In fact, we probably know nothing about the pituitary of the porpoises, ziphiids, we know just a few information about the sperm whale pituitary, and very very few about the mysticete pituitary and do not know whether the differences observed in the present manuscript with respect to the generalized mammalian pituitary may reflect specific adaptations of delphinids or cetaceans in general. For this reason, I recommend the authors to state that their results cannot be generalized to the other cetacean families not only because of the limited number of investigated specimens (as they did on line 839-843) but also because of their extremely limited taxonomic sample.

Two additional comments are necessary:

1.       Simple summary: Line 21: the authors mentioned two species but in the abstract there are three investigated species.

2.       Line 42: replace the full stop after gonadotrophs with a comma

3.       The documents providing the necessary permissions have to be provided maybe as a list with protocol numbers in the Supplementary Information to be sure that ethical concern is not grounded.

Apart from the above comments and recommendations, I think that this is a very interesting and well-done manuscript reflecting a very good research protocol and adequate interpretation of well-described results. For these reasons, I recommend publication with minor revision.

Author Response

Thank you very much for taking the time to review this manuscript. Please find the detailed responses to your insightful comments below, with the corresponding changes marked in red, as well as these corrections in the re-submitted files.

Comment 1: First, I strongly recommend the authors to provide a line drawing of a generalized mammalian hypophysis illustrating the three main portions of this gland. Such an image would be of great help to the non-specialized reader to understand the topological relationships of the different portions of the gland otherwise the jargon terms may represent a barrier to the understanding. Then, I suggest to include in the same image another line drawing with the main characters of the dolphin’s pituitary gland showing the main portions and illuminating the main differences with respect to the generalized mammalian pituitary gland.

Response 1: I strongly agree with this remark, it gives more anatomical context of the structure of the pituitary gland, very useful as we then dicuss thoroughly on many structural elements. We have added a supplementary figure (Figure S1), with a general illustration of the main parts of the mammalian pituitary gland, and then a illustration of that of dolphins. We have referenced it in the main text (Page 2, Paragraph 3, Line 72) "[This differs strikingly from most mammals, where a middle portion (pars intermedia) juxtaposes these structures (Figure S1) [11,12]]"

Comment 2: Second, I recommend the authors to provide line drawings illustrating the different morphological and ultrastructural characters of the cells they described in the text. There is plenty of examples of very nice scientific illustrations of endocrine cells in the literature and adding such an illustration would be of great help for the non-specialized reader as it would provide information that is immediately understandable by everyone as the differences described in the text seem evident. Unfortunately, it is hard to discern peculiar characters of the different cell types from the images obtained by the transmission microscopy and this makes it harder both the reading and understanding.

Response 2:  I agree with this comment and not only find it very insightful but allows a clearer understanding of the manuscript, given the high number of described cells and the technical specificities of electron microscopy, scientific illustrations serve as a general more accesible guide towards the understanding of the text. Moreover, we hope this clarification can help improve the clear presentation of our results.

We have included a Figure in the main text (Figure 8. Schematic representation of the general morphology of the six different hormone-producing endocrine cells in the common dolphin (Delphinus delphis) pituitary gland. (A) Corticotrophs, (B) Lactotrophs, (C) Somatotrophs, (D) Gonadotrophs, (E) Thyrotrophs, and (F) Melanotrophs. The illustration highlights key morphological and ultrastructural features of each cell type as described in the text.Created in BioRender. Alonso, P. (2025) https://BioRender.com/b14i244) illustrating the morphological and ultrastructural specificities of each endocrine adenohypophyseal cell ,  in Page 11, Paragraph 2, Line 389,  and we have referenced it elsewhere in the manuscript's TEM results section were relevant in reference to the text. 

Comment 3: Third, I recommend the authors to emphasize that their results are from three dolphin species (Delphinidae family) and cannot be obviously generalized to the whole Cetacea clade. In fact, we probably know nothing about the pituitary of the porpoises, ziphiids, we know just a few information about the sperm whale pituitary, and very very few about the mysticete pituitary and do not know whether the differences observed in the present manuscript with respect to the generalized mammalian pituitary may reflect specific adaptations of delphinids or cetaceans in general. For this reason, I recommend the authors to state that their results cannot be generalized to the other cetacean families not only because of the limited number of investigated specimens (as they did on line 839-843) but also because of their extremely limited taxonomic sample.

Response 3:

The reviewer raises a very important and valuable point, and until further investigations clear out the basic anatomy and structural components of the pituitary gland of other cetacean species, particularly in Mysticeti species, we need to clearly draw the separation and avoid generalizations. We have stated it clearly that this study gives insight in three dolphin species and reframed all the generalizations and conclusions.

In the discussion section, Page 20, Paragraph 3, Line 710 "[This study provides a comprehensive morphological, histological, and ultrastructural evaluation of the pituitary gland in three dolphin species, introducing novel methodologies to enhance our understanding of this critical endocrine organ Using immunolabeling techniques, we identified specific endocrine cells, including the first reported immunolabeling of thyrotropin-secreting (TSH) cells in the pituitary adenohypophysis of these cetaceans.]" ; subsequently in the rest of the discussion in this matter (changes can be visualized in the attached manuscript), and finally in the discussion section where limitations of the study are discussed, Page 23, Paragraph 7, Line 889 "[

Certain limitations were present in our study, notably the small sample size for electron microscopy, which may constrain the generalizability of our findings. Limited sample sizes may preclude our ability to detect individual or interspecies variations and to assess different physiological states (e.g., seasonal changes, reproductive states, etc.), all of which could influence adenohypophyseal ultrastructure [80–82]. Additionally, our study was restricted to only three species of delphinids, limiting our capacity to extrapolate these findings to the entire cetacean clade. Given the morphological and physiological diversity across cetaceans, further studies encompassing a broader range of taxa are necessary to fully characterize the variability in endocrine cell structure and function within this group.]" .

Moreover, we have already made this specification in the conclusion section, Page 24, 1st Paragraph, Line 917 "[By integrating non-invasive imaging techniques, a novel necropsy protocol, and detailed ultrastructural analyses, we have identified key structural features in three different dolphin species "

Comment 4: Simple summary: Line 21: the authors mentioned two species but in the abstract there are three investigated species

Response 4: Corrected error, simple summary, line 21. "[(...) from three dolphin species (...)]"

Comment 5Line 42: replace the full stop after gonadotrophs with a comma

Response 5: Corrected punctuation error. Line 42 "[corticotrophs (ACTH), somatotrophs (GH), gonadotrophs, (FSH and LH), lactotrophs (LTH), melanotrophs (MSH), thyrotrophs (TSH), follicular cells, and capsular cells.]"

Comment 6:  The documents providing the necessary permissions have to be provided maybe as a list with protocol numbers in the Supplementary Information to be sure that ethical concern is not grounded.

Response 6: We have added the full copy of the authorization into the supplementary materials (Supplementary Document S1) , and we have stated its availability in the main text, Page 3, Paragraph 8, Line 134 "[All necessary permits were obtained from the Spanish Ministry for the Ecological Transition and Demographic Challenge and the Canarian Government’s Environmental Department (project number PID2021-127687NB-10). A full copy of the authorization is included in the Supplementary Materials for reference (Document S1).]"